# Effects of regional limb perfusion technique on concentrations of antibiotic achieved at the target site: A meta-analysis

**Laurel E. Redding**[1], **Elizabeth J. Elzer**[2], **Kyla F. Ortved**[1]*

**1** Department of Clinical Studies, School of Veterinary Medicine, University of Pennsylvania, Kennett Square, Pennsylvania, **2** Rood and Riddle Equine Hospital, Saratoga Springs, New York, United States of America

* kortved@vet.upenn.edu

**Data Availability Statement:** All relevant data are within the paper and its Supporting information files.

**Funding:** The authors received no specific funding for this work.

## Abstract

Intravenous regional limb perfusions (RLP) are widely used in equine medicine to treat distal limb infections, including synovial sepsis. RLPs are generally deemed successful if the peak antibiotic concentration (Cmax) in the sampled synovial structure is at least 8–10 times the minimum inhibitory concentration (MIC) for the bacteria of interest. Despite extensive experimentation and widespread clinical use, the optimal technique for performing a successful perfusion remains unclear. The objective of this meta-analysis was to examine the effect of technique on synovial concentrations of antibiotic and to assess under which conditions Cmax:MIC $\geq$ 10. A literature search including the terms "horse", "equine", and "regional limb perfusion" between 1990 and 2021 was performed. Cmax (µg/ml) and measures of dispersion were extracted from studies and Cmax:MIC was calculated for sensitive and resistant bacteria. Variables included in the analysis included synovial structure sampled, antibiotic dose, tourniquet location, tourniquet duration, general anesthesia versus standing sedation, perfusate volume, tourniquet type, and the concurrent use of local analgesia. Mixed effects meta-regression was performed, and variables significantly associated with the outcome on univariable analysis were added to a multivariable meta-regression model in a step-wise manner. Sensitivity analyses were performed to assess the robustness of our findings. Thirty-six studies with 123 arms (permutations of dose, route, location and timing) were included. Cmax:MIC ranged from 1 to 348 for sensitive bacteria and 0.25 to 87 for resistant bacteria, with mean (SD) time to peak concentration (Tmax) of 29.0 (8.8) minutes. Meta-analyses generated summary values (θ) of 42.8 x MIC and 10.7 x MIC for susceptible and resistant bacteria, respectively, though because of high heterogeneity among studies ($I^2$ = 98.8), these summary variables were not considered reliable. Meta-regression showed that the only variables for which there were statistically significant differences in outcome were the type of tourniquet and the concurrent use of local analgesia: perfusions performed with a wide rubber tourniquet and perfusions performed with the addition of local analgesia achieved significantly greater concentrations of antibiotic. The majority of arms achieved Cmax:MIC $\geq$ 10 for sensitive bacteria but not resistant bacteria. Our results suggest that wide rubber tourniquets and concurrent local analgesia should be strongly considered for

**Competing interests:** The authors have declared that no competing interests exist.

use in RLP and that adequate therapeutic concentrations (Cmax:MIC $\geq$ 10) are often achieved across a variety of techniques for susceptible but not resistant pathogens.

## Introduction

Intravenous regional limb perfusion (RLP) is widely used in equine medicine to treat distal limb infections including synovial sepsis, osteomyelitis and cellulitis [1–3]. This technique is an accepted means of delivering high concentrations of antibiotics via the peripheral vasculature to regions of the limb isolated by a tourniquet [4]. It is proposed that the injection of the antibiotic increases local venous pressure and creates a concentration gradient both of which drive the antibiotic into peripheral tissues [5, 6]. One of the major advantages of RLP is the ability to achieve high concentrations of antibiotics in local tissues while maintaining low systemic concentrations [6]. This allows otherwise toxic and prohibitively expensive antibiotics to be used when indicated. Despite the widespread clinical use of regional limb perfusion in the horse, the optimal technique remains unclear.

A plethora of experimental studies evaluating different RLP techniques have been performed over the past two decades. In most experimental studies, the success of RLP is measured by quantifying the peak antibiotic concentration (Cmax) achieved in the synovial fluid of selected synovial structures and calculating the ratio of Cmax to the minimum inhibitory concentration (MIC), determined for the antibiotic and bacteria of interest. A Cmax:MIC of at least 8:1 to 10:1 has been recommended for aminoglycosides to be effective [7]. As recently reviewed by Biasutti et al. (2021), commonly investigated factors affecting the success of RLP include tourniquet type, tourniquet duration, tourniquet location in relation to the synovial structure sampled, perfusate volume, the use of general and regional anesthesia, and antibiotic dose [4]. Agreement among studies on the effects of each factor varies.

Meta-analysis allows statistical combination of results of multiple studies, weighted according to sample size and study precision. This technique increases statistical power and provides evidence for clinical decision making and future research needs [8]. The volume and variability of results in the RLP literature provides an opportunity for meta-analytic synthesis. The objective of this meta-analysis was to evaluate the effect of RLP technique on synovial concentrations of antibiotic and to assess under which conditions Cmax:MIC $\geq$ 10. Variables examined included tourniquet type, tourniquet duration, general anesthesia versus standing sedation, perfusate volume, antibiotic dose, synovial structure sampled, and the concurrent use of local analgesia.

## Materials and methods

This meta-analysis was conducted according the recommendations of the PRISMA statement [8]. A protocol for Systematic Review and registration was not required.

### Literature search

A literature search was carried out using PubMed/MEDLINE for date range January 1990 to December 2021. Search terms used included "horse", "equine", "regional limb perfusion". Secondary searches were performed using the search engines of veterinary trade journals. Additional papers were identified in the bibliographies of relevant articles. Studies were included if they 1) involved equine species, 2) consisted of prospective, randomized studies comparing the effects of two or more techniques for intravenous RLP or pharmacokinetic studies of

intravenous RLP of individual antibiotics; and 3) measured synovial concentration of the antibiotic at the time of tourniquet removal. Studies of intra-osseous RLP, RLP with two antibiotics mixed in one syringe, RLP using substances other than antibiotics, and studies that measured antibiotic concentrations in tissues other than synovial fluid were excluded. Studies examining clinical outcome (e.g., survival, return to use) and not synovial antibiotic concentration were also excluded. Any study for which it was not possible to extract or obtain the required data was excluded. Individual studies were evaluated for risk of bias with respect to randomization, incomplete reporting of outcome data, and selective outcome reporting.

## Outcomes assessed

Because the most commonly cited therapeutic target for RLP is Cmax:MIC $\geq$ 10 for the most common equine orthopedic pathogens, the primary outcome measure for this study was Cmax:MIC. Since the most commonly isolated equine orthopedic pathogens are *Staphylococcus aureus* and *Enterobacteriaceae* [9–11], MICs for susceptible and resistant strains of these bacteria were obtained from the Clinical and Laboratory Standards Institute (CLSI) [12]. The mean peak concentration of antibiotic and the standard errors were then transformed to multiples of the minimum inhibitory concentration (MIC) for susceptible ($MIC_{sus}$) and resistant organisms ($MIC_{res}$).

## Data extraction

Identification of eligible articles against predetermined inclusion and exclusion criteria, and extraction of data was performed by blindly by two authors (EJE and KFO). Discrepancies were resolved by consensus. Data collected for purposes of uniquely identifying studies were article title, authors' names, journal name, year of publication, volume, and page numbers. The outcome extracted from all studies was the mean peak concentration (Cmax) of antibiotic in µg/ml and measures of dispersion. Because standard errors of the outcome are required as input for a meta-analysis, we estimated these values for the arms where they were not provided in the original paper. In studies where only a standard deviation was given, the standard error (SE) was calculated as:

$$SE = \frac{SD}{\sqrt{n}}$$

where SD is the standard deviation and n is the number of horses in the arim.

When a standard deviation was missing but a 95% confidence interval was provided (n = 13 arms), the standard deviation was estimated as: [13]

$$SD = \sqrt{n} * \frac{upper\ limit\ of\ CI - lower\ limit\ of\ CI}{3.92}$$

When a standard deviation was missing but a range of values was provided (n = 54 arms), the standard deviation was estimated as:

$$SD = \frac{upper\ limit\ of\ range - lower\ limit\ of\ range}{\xi(n)}$$

Where $\xi(n) = 2 * E(Z_n)$ and $E(Z_n)$ is the expected value of the Z-statistic for the sample size n. [14] Values for $\xi(n)$ are provided as supplementary material, as reported by Wan et al. [14]

In a small number of case (n = 6 arms), only an interquartile range (IQR) for the outcome was provided. In those cases, SD was calculated as:

$$SD = \frac{upper\ limit\ of\ IQR - lower\ limit\ of\ IQR}{\eta(n)}$$

where $\eta(n) = 2^*E(Z_{(3Q+1)})$ and Q = (n-1)/4. Values for $\eta(n)$ are provided as supplementary material, as reported by Wan et al. [14]. When measures of dispersion were not provided in the text of the manuscript (2 manuscripts, 5 arms), these values were extracted from manuscript figures using WebPlotDigitizer (v.4.3) [15].

Other data extracted from the studies included the number of horses perfused, tourniquet type and pressure (for pneumatic tourniquets), tourniquet location, antibiotic used, perfusate volume, synovial structure sampled, times of sampling, peak antibiotic concentration achieved, and time of peak antibiotic concentration. Tourniquet type was divided into three categories (wide rubber, pneumatic with $\leq$ 400 mmHg pressure, pneumatic with $>$ 400 mmHg pressure). Tourniquet time was divided into two categories ($\leq$ 25 minutes and $>$ 25 minutes). Perfusate volume was divided into three categories (6–30 mL, 40–60 mL, and 100–120 mL). Antibiotic dose was divided into three categories ($<$ 1/3 systemic dose, 1/3-2/3 systemic dose, and $>$ 2/3 systemic dose). An interaction term for structure sampled and tourniquet number and location was also generated, where "near" denoted structures sampled near the distal limit of perfusion (i.e., the distal interphalangeal joint, the metacarpophalangeal joint, and any carpal or tarsal joint isolated by a second distal tourniquet) and "far" denoted structures sampled remotely from the distal limit of perfusion (i.e., any carpal or tarsal joint not isolated by a second distal tourniquet).

## Data analysis

Because most studies reported multiple RLP arms with varying parameters, we considered each arm to be a unique study. However, to account for clustering of arms within studies, we performed random-effects meta-analyses for each outcome (multiples of MIC for susceptible organisms and resistant organisms). Heterogeneity between studies was assessed by calculating $I^2$ and Cochran's Q statistics following the meta-analysis. Briefly, $I^2$ provides an estimate of the percentage of variability in results across studies that is due to real differences and not due to chance, while the Q statistic tests the null hypothesis that the summary measure is the same across studies and that variations are simply caused by chance.

To further explore sources of heterogeneity among studies and their effects on the outcome, mixed effects meta-regression was performed. Variables significantly associated with the outcome on univariable meta-regression (p<0.10) were added to a multivariable meta-regression model. Backwards elimination was then performed in a stepwise manner to identify the parameters that remained statistically significantly associated with the outcome, using a Bonferroni-corrected p-value of 0.006. Forest plots were generated to visualize the results of the meta-regression and identify situations in which Cmax:MIC $\geq$ 10. Meta-analysis and meta-regression were performed using the "meta" package in Stata v.16.0 (StataCorp, College Station, TX). Postestimation regression diagnostics were performed using the "metapred" package in Stata, including verification of normality of residuals, identification of outlying values using standardized residuals, and detection of influencing values using Cook's distance.

## Sensitivity analyses

Because measures of dispersion were not always provided in all studies, standard deviations frequently needed to be estimated. The formulae used to estimate standard deviation require

certain assumptions, such as a normal distribution of results, that may not always be met. We therefore performed sensitivity analyses to compare results of the meta-regression in all studies and in only the studies where standard deviations/errors were extracted directly from the study rather than estimated. Because there was some variation in the type of antibiotic used across different studies, we also performed sensitivity analyses to compare results of the meta-regression in all studies and in studies of amikacin (the most frequently used antibiotic) only. Finally, we repeated meta-regression excluding arms identified as being outliers or having out-sized influence on the results of the model.

## Results

### Description of included studies

The above search strategy identified 158 peer-reviewed publications. After the full application of inclusion criteria, 39 studies remained (Fig 1). No studies were excluded due to excessive risk of bias. During preliminary data gathering, one study with a Cmax:MIC value of 1,571 was deemed to be an outlier and was excluded [16].

Thirty-six studies were deemed appropriate for meta-analysis. Characteristics of the included studies and relevant MICs from the CLSI are described in Table 1 and S1 Table. Overall, studies contained measured outcomes for between 1 and 8 different combinations of

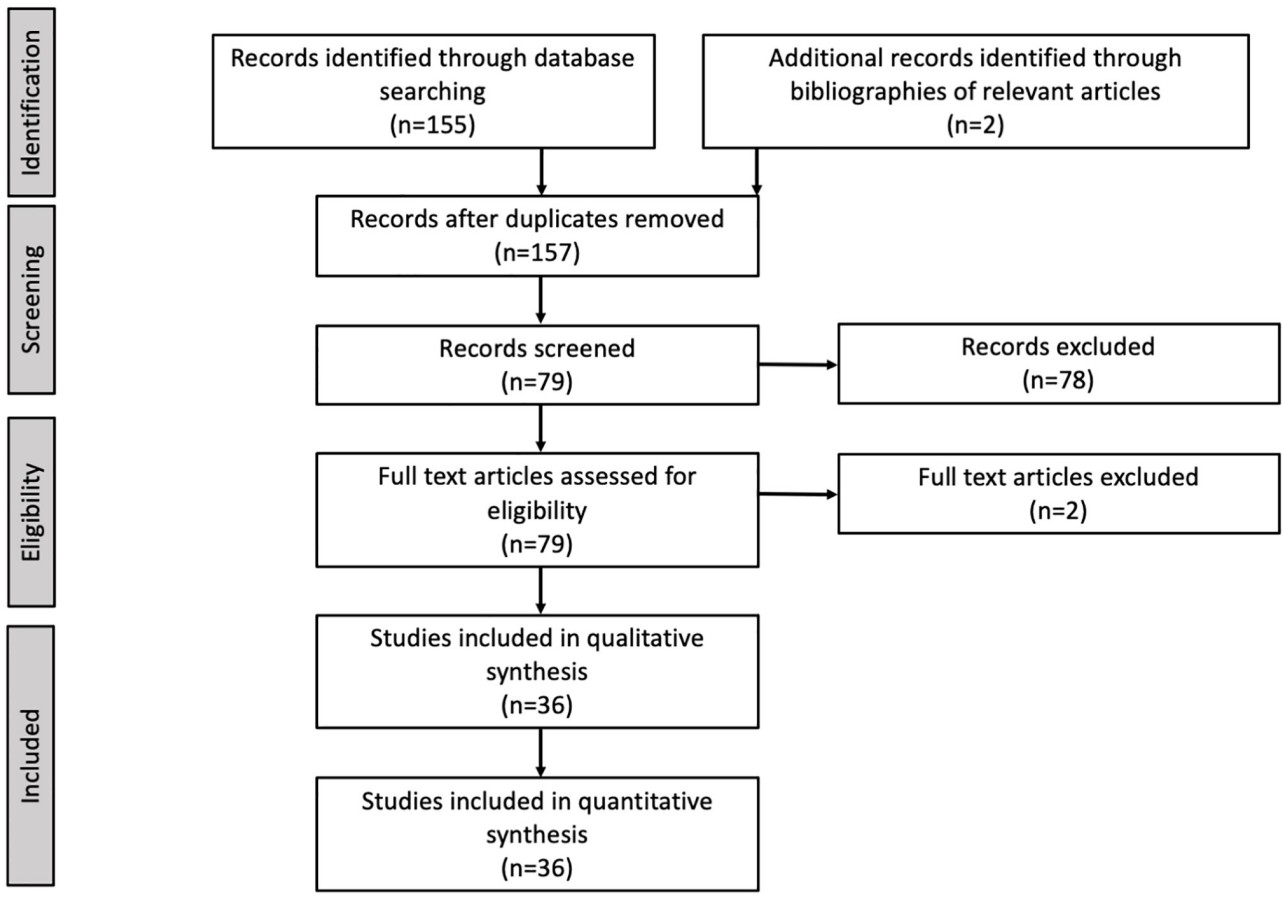

**Fig 1. Flow diagram of included studies.**

**Table 1. Study characteristics for included studies.**

| Study Number | First Author (year) | Number of Arms | Parameters Varied | Number of Horses | Antimicrobial | Measure of Dispersion |
|---|---|---|---|---|---|---|
| 1 | Murphey (1999) [17] | 4 | • Synovial structure sampled<br>• Tourniquet time | 8 | Amikacin | Standard error |
| 2 | Butt (2001) [18] | 3 | • Synovial structure sampled | 6 | Amikacin | Standard deviation |
| 3 | Scheuch (2002) [19] | 1 | | 5 | Amikacin | Standard error |
| 4 | Rubio-Martinez (2005) [20] | 6 | • Synovial structure sampled<br>• Tourniquet time | 6 | Vancomycin | Range |
| 5 | Parra-Sanchez (2006) [21] | 2 | • Antimicrobial | 7 | Amikacin Enrofloxacin | Range |
| 6 | Errico (2008) [22] | 2 | • Synovial structure sampled | 6 | Amikacin | Standard deviation |
| 7 | Levine (2010) [23] | 2 | • Tourniquet type | 6 | Amikacin | Range |
| 8 | Alkabes (2011) [24] | 2 | • Tourniquet type | 6 | Amikacin | Range |
| 9 | Beccar-Varela (2011) [25] | 1 | | 8 | Amikacin | Range |
| 10 | Hyde (2013) [26] | 3 | • Volume | 6 | Gentamicin | Range |
| 11 | Kelmer (2013) [27] | 2 | • Synovial structure sampled | 6 | Erythromycin | 95% Confidence interval |
| 12 | Kelmer (2013) [28] | 3 | • Volume<br>• Vessel<br>• Dose | 6 | Amikacin | 95% Confidence interval |
| 13 | Lallemand (2013) [29] | 1 | | 6 | Marbofloxacin | Standard deviation |
| 14 | Mahne (2014) [30] | 4 | • General anesthesia<br>• Local analgesia | 8 | Amikacin | Standard deviation |
| 15 | Zantingh (2014) [31] | 1 | | 6 | Amikacin | Interquartile range |
| 16 | Kelmer (2015) [32] | 2 | • Synovial structure sampled | 4 | Chloramphenicol | Standard deviation |
| 17 | Sole (2015) [33] | 4 | • Synovial structure sampled<br>• Tourniquet location | 8 | Amikacin | Standard deviation |
| 18 | Aristizabal (2016) [34] | 8 | • Synovial structure sampled<br>• Tourniquet location<br>• Tourniquet time | 6 | Amikacin | Standard deviation |
| 19 | Colbath (2016) [35] | 2 | • Local analgesia | 7 | Amikacin | Interquartile range |
| 20 | Godfrey (2016) [36] | 2 | • Volume | 8 | Amikacin | 95% Confidence interval |
| 21 | Harvey (2016) [37] | 4 | • Dose<br>• Torniquet location | 6 | Amikacin | Range |
| 22 | Kilcoyne (2016) [38] | 4 | • Synovial structure sampled<br>• Tourniquet location | 7 | Amikacin | Standard deviation |
| 23 | Moser (2016) [5] | 16 | • Volume<br>• Synovial structure sampled<br>• Tourniquet location<br>• Tourniquet time | 6 | Amikacin | 95% Confidence interval |
| 24 | Oreff (2016) [39] | 3 | • Volume | 7 | Amikacin | Range |
| 25 | Dahan (2017) [40] | 1 | | 6 | Imipenem | Interquartile range |

*(Continued)*

**Table 1.** (Continued)

| Study Number | First Author (year) | Number of Arms | Parameters Varied | Number of Horses | Antimicrobial | Measure of Dispersion |
|---|---|---|---|---|---|---|
| 26 | Kelmer (2017) [41] | 2 | • Synovial structure sampled | 5 | Imipenem | Standard deviation |
| 27 | Oreff (2017) [42] | 1 | | 8 | Ceftazidime | Range |
| 28 | Schoonover (2017) [43] | 12 | • Tourniquet location<br>• Synovial structure sampled<br>• Tourniquet time | 6 | Amikacin | 95% CI |
| 29 | Fontenot (2018) [44] | 1 | | 9 | Meropenem | Standard deviation |
| 30 | Kilcoyne (2018) [45] | 4 | • Tourniquet location<br>• Tourniquet time | 7 | Amikacin | Standard deviation |
| 31 | Dahan (2019) [46] | 1 | | 6 | Amikacin | Interquartile range |
| 32 | Snowden (2019) [47] | 3 | • Tourniquet type<br>• Synovial structure sampled | | Polymixin B | Range |
| 33 | Gustafsson (2020) [48] | 6 | • Tourniquet time | 6 | Amikacin | Standard deviation |
| 34 | Gustafsson (2021) [49] | 2 | • Antimicrobial | 10 | Trimethoprim Sulfadiazine | Interquartile range |
| 35 | Gustafsson (2021) [50] | 2 | • Tourniquet time | 11 | Metronidazole | Standard deviation |
| 36 | Kilcoyne (2021) [51] | 6 | • Tourniquet time | 7 | Amikacin | Range |

perfusion technique, for a total of 123 permutations of perfusion technique (or "arms") among all studies. For one study (Kelmer et al. 2013), we were unable to obtain measures of dispersion for one of the arms, therefore the final number of arms in the meta-analysis was 122.

The outcome variable Cmax:MIC ranged from 1 to 348 for susceptible organisms, and from 0.25 to 87 for resistant organisms, and the mean (SD) time to peak concentration (Tmax) was 29.0 (8.8) minutes. Meta-analyses generated summary values ($\theta$) of 42.8 x MIC and 10.7 x MIC for susceptible and resistant organisms, respectively. However, the heterogeneity in study designs and perfusion parameters (drug, timing, synovial structure, etc.) across studies was high. $I^2$, which represents the percentage of residual between-study variation relative to the total variability, was equal to 98.8, which reflects large heterogeneity (S1 and S2 Figs) [52]. As a result, summary measures were not considered reliable.

On univariable meta-regression, three variables were statistically significantly associated with the outcome: volume of the perfusate, the type of tourniquet used, and addition of local analgesia. High volumes of perfusate (100–120 mL) resulted in significantly higher (23.4 x MIC, p = 0.014) concentrations than low volumes (30 mL or less). Concentrations were not statistically significantly different for medium (40–60 mL) and low volumes.

Pneumatic tourniquets achieved lower concentrations than wide rubber tourniquets, with pressures greater and lower than 400 mm Hg achieving 29.4 and 1.74 fewer MICs, respectively. However, this effect was only statistically significant for pneumatic tourniquets with pressures greater than 400 mm Hg (p<0.001).

Perfusates with added analgesia–either via the addition of mepivacaine to the infusion or the administration of a nerve block before the infusion—achieved significantly higher levels of antibiotics (32.8 x MIC, p<0.001) than perfusions without analgesia.

On multivariable meta-regression, the volume of perfusate was no longer significantly associated with the outcome. Pneumatic tourniquets with pressures greater than 400 mmHg achieved significantly lower concentrations (20.3 x MIC fewer, p = 0.006) than wide rubber

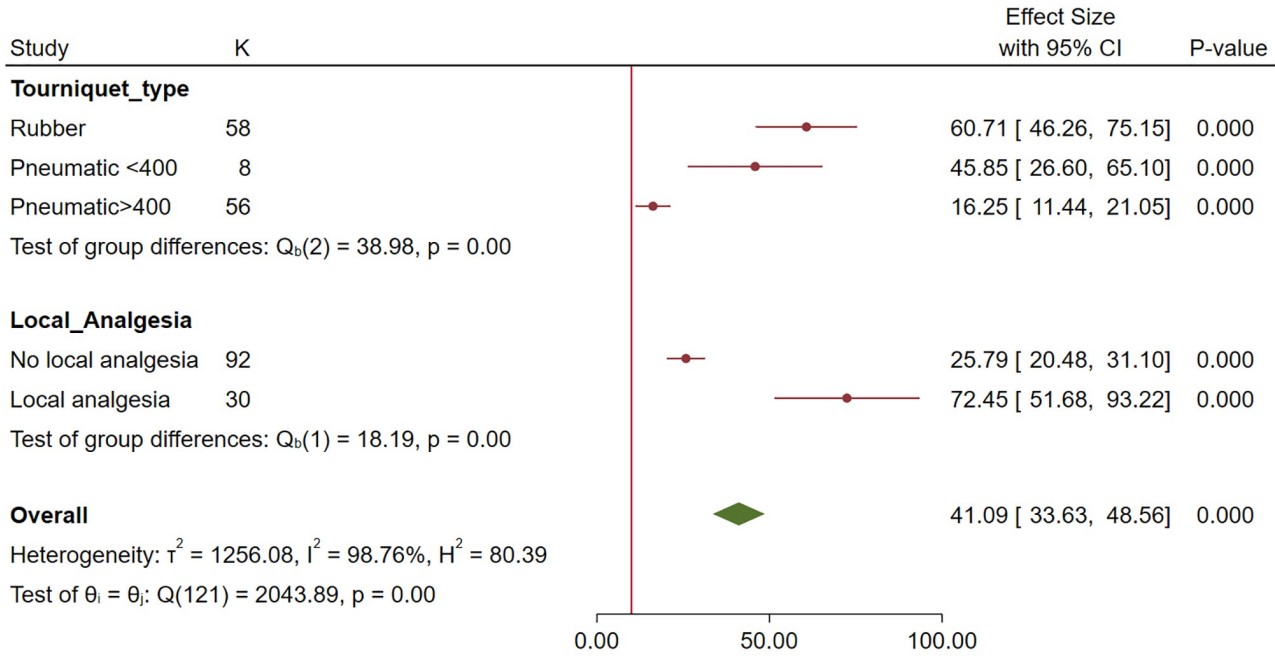

**Fig 2. Forest plot by subgroups retained in the final multivariable meta-regression showing concentrations of antibiotics achieved in the synovial structure (multiples of the MIC) for susceptible bacteria.** The vertical red line represents 10x MIC. Red circles represent point estimates, red horizontal lines represent 95% confidence intervals. The green diamond represents the summary measure θ, but this measure is not considered reliable due to the high heterogeneity between studies.

tourniquets, while perfusates with added analgesia achieved greater concentrations (22.4 x MIC more, p<0.004). As shown in Fig 2, for susceptible bacteria, more than 10x MIC was achieved for all subgroups. However, as shown in Fig 3, for resistant bacteria, more than 10 x MIC was achieved for only rubber tourniquets and perfusates with added analgesia.

Finally, meta-analysis showed that synovial structure, % systemic dose, tourniquet location, general anesthesia vs. standing, and tourniquet time were not statistically significantly associated with the outcome (Figs 4 and 5).

## Sensitivity analyses

Sensitivity analyses (Table 2) showed that similar trends and degrees of significance were achieved for meta-regression, regardless of the groups of arms used. Coefficients tended to be biased towards the null when considering all arms relative to the above-mentioned subgroups of arms.

## Discussion

While other studies have qualitatively examined the effect of variability in perfusion parameters on levels of antibiotics achieved in the synovial structure, this was the first study to do so quantitatively using meta-analytic methods. This meta-analysis combined data from 36 studies and 123 arms to evaluate the effects of several variables on synovial concentrations of antibiotic. Interestingly, upon univariable meta-regression, only perfusate volume, tourniquet type and the concurrent use of local analgesia significantly affected the outcome, while tourniquet

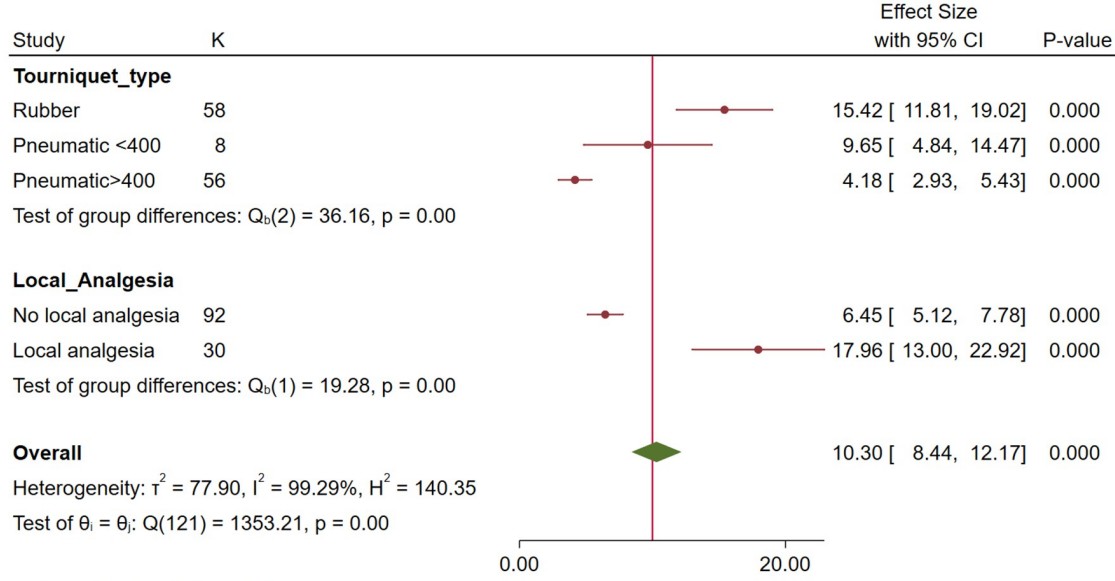

**Fig 3. Forest plot by subgroups retained in the final multivariable meta-regression showing concentrations of antibiotics achieved in the synovial structure (multiples of the MIC) for resistant bacteria.** Red circles represent point estimates, red horizontal lines represent 95% confidence intervals. The green diamond represents the summary measure θ, but this measure is not considered reliable due to the high heterogeneity between studies.

type and concurrent use of local analgesia were the only two variables retained in the final multivariable meta-regression.

Although volume of perfusate was not significantly associated with the outcome in the final multivariable analysis, we did find that higher volumes of perfusate achieved higher levels of Cmax than lower levels. Larger volumes likely achieve higher intravascular pressures facilitating diffusion of the antibiotic into peripheral tissues. When RLP is used clinically it is generally repeated daily for varying periods of time i.e., days to weeks. Therefore, the potential complications of repeated large volume perfusions, including vasculitis, when RLP is used clinically, must be considered. Selecting the lowest perfusate volume that reliably achieves an efficacious Cmax:MIC would be clinically beneficial. From this meta-analysis, low, medium and high volume perfusions achieved Cmax:MIC $\geq$ 10 for sensitive bacteria, while only high volume perfusions achieved Cmax:MIC $\geq$ 10 for resistant bacteria. Therefore, clinicians are encouraged to consider the bacterial sensitivity when selecting their RLP technique.

The final model demonstrated that tourniquet type and the concurrent use of local analgesia were the two most important variables associated with synovial concentrations of antibiotic. Our meta-analysis showed that pneumatic tourniquets were associated with lower antibiotic concentrations than wide rubber tourniquets and that pneumatic tourniquets with pressures > 400mm Hg were the least successful at achieving Cmax:MIC $\geq$ 10. One suggested benefit of pneumatic tourniquets is the ability to standardize pressure between horses, while benefits of rubber tourniquets include less risk of pressure loss associated with leakage or bursting of pneumatic tourniquets. Clinically, wide rubber tourniquets are economical, easy to use and accessible, and their efficacy is clearly supported by the current body of literature.

Regional limb perfusion was first performed under general anesthesia to limit the animal's movement. However, the clinical application of RLP is much simpler and more accessible

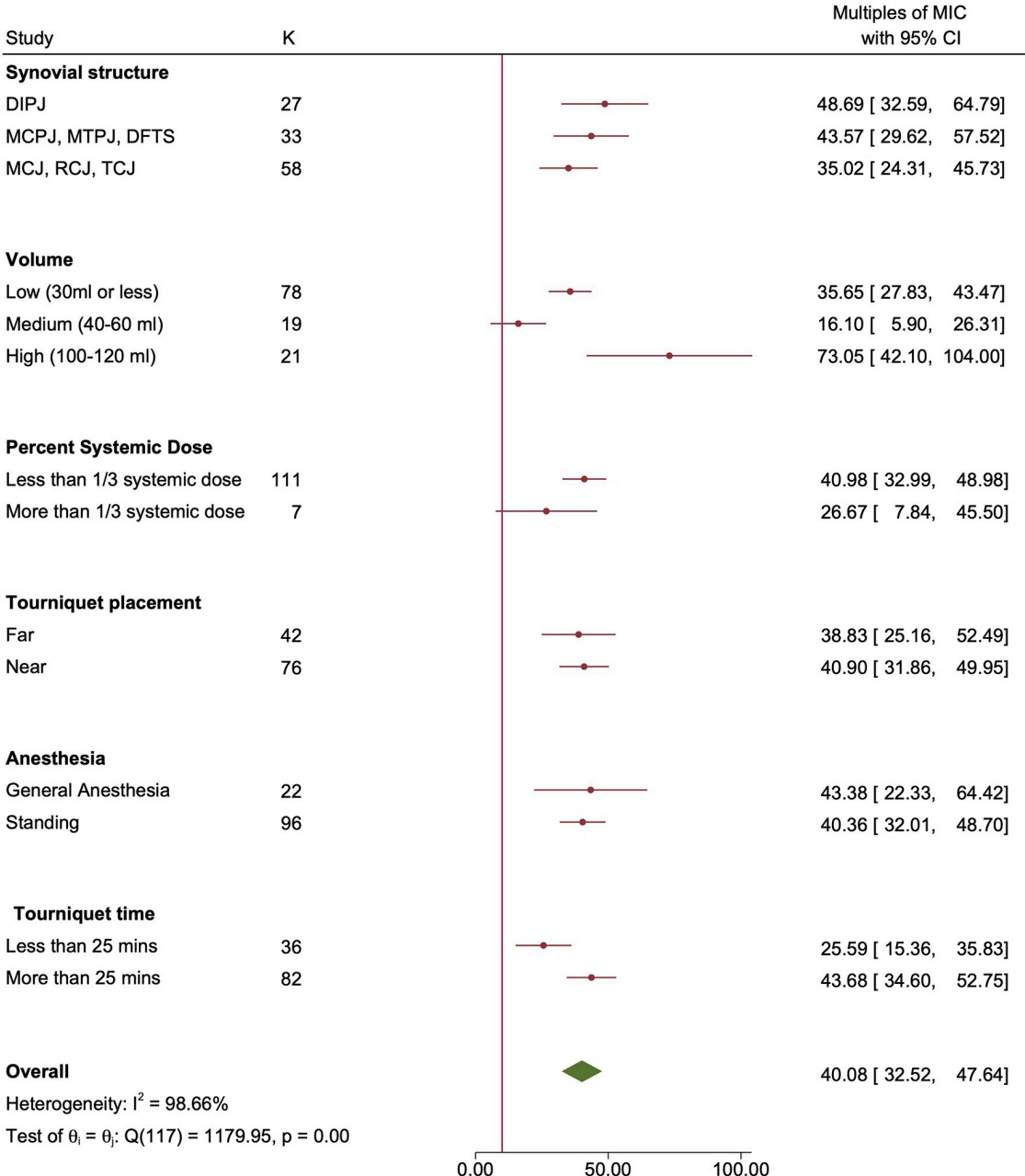

**Fig 4. Forest plot by subgroups with variables for which there were no statistically significant differences in outcome.** The concentrations of antibiotics achieved in the synovial structure (multiples of the MIC) for susceptible bacteria are shown. The vertical red line represents 10x MIC. Red circles represent point estimates, red horizontal lines represent 95% confidence intervals. The green diamond represents the summary measure θ, but this measure is not considered reliable due to the high heterogeneity between studies.

using standing sedation, and RLPs performed under general anesthesia and standing sedation achieve similar synovial concentrations of antibiotics [30, 34, 35]. Nevertheless, horse movement under standing sedation has been cited as a plausible reason for perfusion failure [23, 35]. In the present study, we found that the concurrent use of local analgesia, either via locoregional nerve blocks or the addition of a local anesthetic to the perfusate, was strongly and

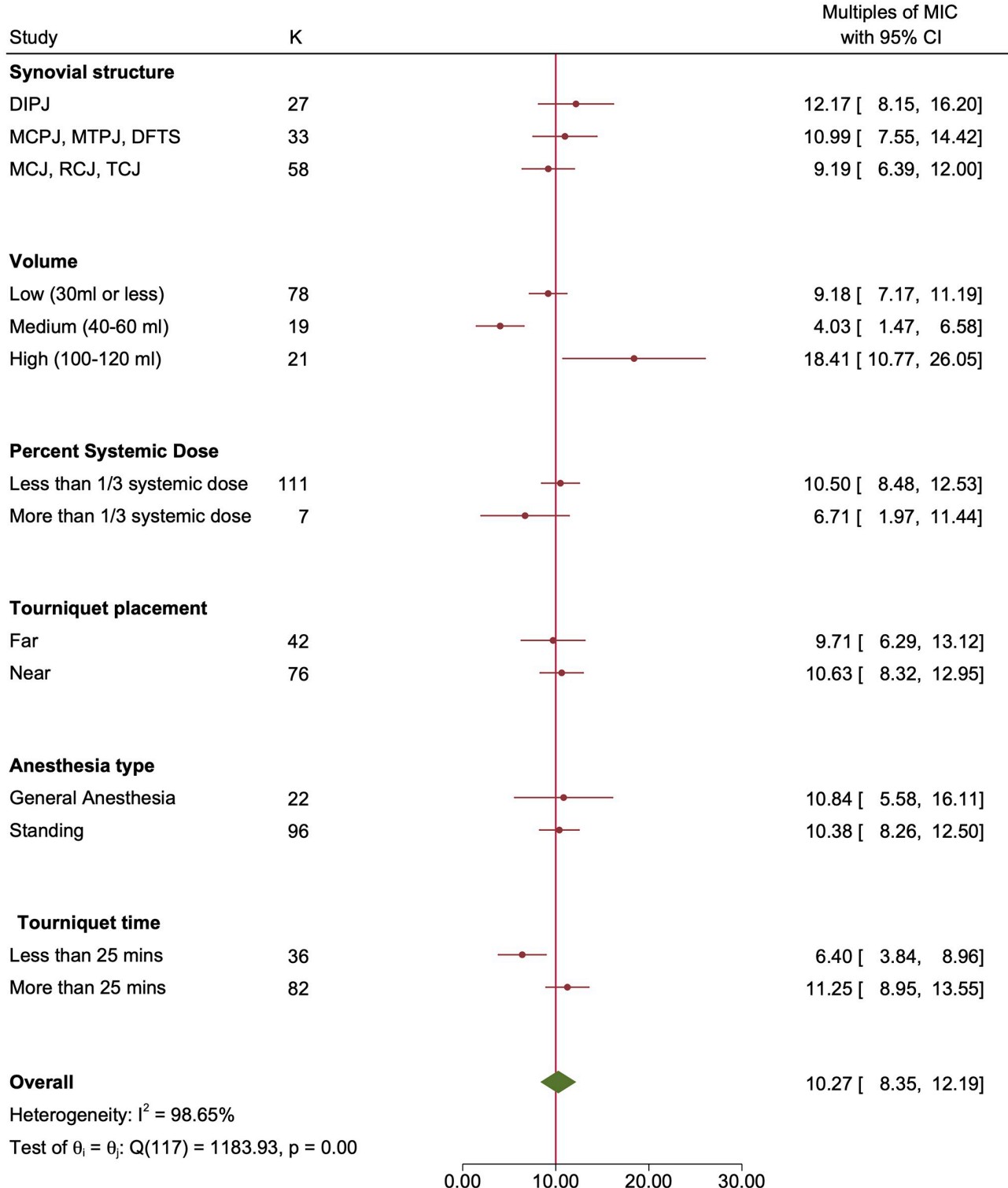

**Fig 5. Forest plot by subgroups with variables for which there were no statistically significant differences in outcome.** The concentrations of antibiotics achieved in the synovial structure (multiples of the MIC) for resistant bacteria are shown. The vertical red line represents 10x MIC. Red circles represent point estimates, red horizontal lines represent 95% confidence intervals. The green diamond represents the summary measure θ, but this measure is not considered reliable due to the high heterogeneity between studies.

**Table 2. Sensitivity analyses showed that similar trends and degrees of significance were achieved for meta-regression, regardless of the groups used.**

| | All arms (n = 122) | | | Arms where standard deviations not estimated from range (n = 64) | | | Arms of amikacin only (n = 95) | | |
|---|---|---|---|---|---|---|---|---|---|
| Variable | Coefficient | P-value | 95% CI | Coefficient | P-value | 95% CI | Coefficient | P-value | 95% CI |
| **Tourniquet type** | | | | | | | | | |
| • Rubber | [Ref] | [Ref] | [Ref] | [Ref] | [Ref] | [Ref] | [Ref] | [Ref] | [Ref] |
| • Pneumatic <400 mm Hg | 3.72 | 0.789 | -23.4–30.9 | 5.97 | 0.790 | -38.1–50.0 | 3.36 | 0.883 | -41.4–48.1 |
| • Pneumatic >400 mm HG | -20.3 | 0.006 | -34.9-(-5.69) | -22.9 | 0.102 | -50.5–4.53 | -25.7 | 0.008 | -44.7-(-6.71) |
| Local analgesia | | | | | | | | | |
| • None | [Ref] | [Ref] | [Ref] | [Ref] | [Ref] | [Ref] | [Ref] | [Ref] | [Ref] |
| • Local analgesia | 24.0 | 0.004 | 7.80–40.2 | 38.7 | 0.005 | 11.5–66.0 | 37.8 | 0.001 | 14.5–61.0 |

Coefficients tended to be biased towards the null when considering all arms relative to subgroups of arms.

significantly associated with achieving high Cmax in synovial structures. Tourniquet-associated pain is a reported complication of use in human medicine [53]. Concurrent use of analgesia alongside RLP appears useful in limiting animal discomfort and subsequent movement, thus decreasing the risk of tourniquet failure. Errico et al. (2008) [22], based on preliminary studies, also suggested that weight shifting due to tourniquet associated pain leads to doubling of the intravascular pressure distal to the tourniquet which can lead to systemic leakage. Based on this meta-analysis, concurrent analgesia appears to be particularly important when treating resistant bacterial infections.

Other parameters that were not found to be statistically significantly associated with higher synovial concentrations of antibiotic included the duration of the perfusion, where the tourniquet was placed relative to the synovial structure (i.e., near or far), whether the horse was standing or under general anesthesia, and the dose of antibiotic (percent of systemic dose). Therapeutic concentrations of antibiotics have been achieved in synovial fluid in 10 minutes [38], 15 minutes [2], 20 minutes [34], 30 minutes [17], and 45 minutes after tourniquet placement [20]. Although we found that the mean time to peak concentration was 29.0 minutes, Cmax:MIC ≥ 10 is reached earlier. Because studies only collect synovial fluid at predetermined time points, it is not possible to determine exactly when Cmax:MIC ≥ 10. Only 16 arms in 5 studies examined synovial fluid antibiotic concentration at time < 30 minutes. These included 2 arms that examined concentrations at 10 minutes, 12 that examined concentrations at 20 minutes, and 2 that examined concentrations at 25 minutes. Of these arms, 12/16 achieved Cmax:MIC ≥ 10 for sensitive bacteria and 6/16 achieved Cmax:MIC ≥ 10 for resistant bacteria (S1 Table). When using RLP to treat clinical infections, leaving a tourniquet in place for 30 minutes may be advisable when feasible to achieve the highest Cmax possible. However, individual horse behavior, movement, and response to sedation during RLP can be unpredictable, and shorter perfusate times may be necessary.

Several experiments have measured synovial fluid Cmax in multiple synovial structures to evaluate the effect of distance from tourniquet to synovial structure on antibiotic concentration. Although some researchers have hypothesized that Cmax will be lower in synovial structures that are further away from the tourniquet, studies have shown that a proximal tourniquet with perfusion into the cephalic or saphenous vein can produce high levels of antibiotics in distant synovial structures such as the distal interphalangeal joint [5, 43]. The results of our meta-analysis suggest that high concentrations of antibiotics are achieved in synovial structures regardless of tourniquet location. This is advantageous, as horses can be more reactive to needle placement in palmar/plantar digital veins compared to cephalic or saphenous veins.

Additionally, this may allow clinicians to begin regional limb perfusions in a more proximal location, allowing for more distal locations to be used in sequential treatments if there are complications such as phlebitis.

Therapeutic concentrations of antibiotic in synovial fluid have been achieved using doses ranging from 2% [17] to 91% [16] of the recommended systemic dose. Some researchers and clinicians have recommended performing regional limb perfusion with 1/3 of the systemic dose of the desired antibiotic [25], while others have suggested administering the full systemic dose via the RLP without concurrent systemic administration [54]. The lack of statistical significance of dose on Cmax:MIC in this meta-analysis may be influenced by the small number of studies evaluating high doses of antibiotics. Of the 123 arms, 115 arms used < 1/3 of the systemic dose of the antibiotic. From this data, administering 1/3 of the systemic dose can be considered efficacious for susceptible bacteria and that administration of high doses is not necessary to achieve Cmax:MIC $\geq$ 10. However, treatment of infections with resistant bacteria may require higher doses to achieve an effective Cmax:MIC.

Limitations of this study are those typically associated with meta-analyses, most notably the presence of heterogeneity among the studies. We sought to explore heterogeneity by performing meta-regression, as recommended by the Cochrane group [55], but we were ultimately unable to generate trustworthy summary values for Cmax across all conditions. A benefit of our meta-analysis is that it is unlikely that there was significant publication bias among considered studies, as our outcome was a continuous value (concentration of antibiotic) rather than a categorical success/failure outcome for which negative findings might not be published. The inclusion of experimental studies only rather than observational studies or randomized controlled trials also means that issues of randomization, bias, and dropout, which are typically used as markers of study quality in meta-analyses [56, 57], are not applicable here. Moreover, since the trends in our findings persisted with sensitivity analyses, we believe our findings to be relatively robust.

## Conclusions

As expected, the considerable heterogeneity in technique between studies precluded reporting of an overall summary measure for Cmax:MIC for RLP. However, our meta-regressions allowed us to narrow in on the parameters that most affected synovial concentrations of antibiotic, the most important of which appear to be the type of tourniquet and the delivery of analgesia along with the antibiotic. Our results suggest that wide rubber tourniquets and concurrent local analgesia should be strongly considered for use in RLP. Our results also indicate that adequate therapeutic concentrations ($\geq$10x MIC) can often be achieved across a variety of techniques for susceptible but not resistant pathogens.

## Supporting information

**S1 Fig. Forest plot of all studies included in the meta-analysis showing Cmax:MIC for susceptible bacteria.** A vertical red line depicts 10x MIC.
(PDF)

**S2 Fig. Forest plot of all studies included in the meta-analysis showing Cmax:MIC for resistant bacteria.** A vertical red line depicts 10x MIC.
(PDF)

**S1 Table. Study characteristics including Cmax, measures of dispersion and calculated Cmax:MIC for susceptible and resistant bacteria for included arms.** lpdv = lateral palmar digital vein; saph = saphenous; ceph = cephalic; DIPJ = distal interphalangeal joint;

MCPJ = metacarphophalangeal joint; TCJ = tarsocrural joint; RCJ = radiocarpal joint; MCJ = middle carpal joint; DFTS = digital flexor tendon sheath.
(XLSX)

**S2 Table. Values of $\xi$ (n) in the formula (7) and the formula (12) for Q $\leq$ 20.** Modified from Wan et al. 2014. [14].
(DOCX)

**S3 Table. Values of $\eta$ (n) in the formula (12) and the formula (15) for Q $\leq$ 20, where n = 4Q + 1.** Modified from Wan et al. 2014. [14].
(DOCX)

**S1 Checklist. PRISMA 2020 checklist.**
(DOCX)

## Author Contributions

**Conceptualization:** Laurel E. Redding, Elizabeth J. Elzer.

**Data curation:** Laurel E. Redding, Elizabeth J. Elzer, Kyla F. Ortved.

**Formal analysis:** Laurel E. Redding, Kyla F. Ortved.

**Methodology:** Laurel E. Redding.

**Supervision:** Kyla F. Ortved.

**Writing – original draft:** Laurel E. Redding, Elizabeth J. Elzer, Kyla F. Ortved.

**Writing – review & editing:** Laurel E. Redding, Elizabeth J. Elzer, Kyla F. Ortved.

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
