## [Decision Letter · Decision Letter 0]

25 Feb 2022

PONE-D-22-00403Effects of regional limb perfusion technique on concentrations of antibiotic achieved at the target site: a meta-analysisPLOS ONE

Dear Dr. Ortved,

Thank you for submitting your manuscript to PLOS ONE. After careful consideration, we feel that it has merit but does not fully meet PLOS ONE’s publication criteria as it currently stands. Therefore, we invite you to submit a revised version of the manuscript that addresses the points raised during the review process.

Thank you for writing a concise, well-organized manuscript--that makes it easier to review. The reviewers tried hard to ask clarifying questions and suggest edits without excessive nit-picking. The most challenging aspect in the re-writing will be addressing the Type I error inflation issue.

We look forward to receiving your revised manuscript.

Kind regards,

Richard Evans

Academic Editor

PLOS ONE

Journal Requirements:

Reviewers' comments:

Reviewer's Responses to Questions

**Comments to the Author**

1. Is the manuscript technically sound, and do the data support the conclusions?

Reviewer #1: Yes

Reviewer #2: Yes

2. Has the statistical analysis been performed appropriately and rigorously? 

Reviewer #1: Yes

Reviewer #2: Yes

3. Have the authors made all data underlying the findings in their manuscript fully available?

Reviewer #1: Yes

Reviewer #2: Yes

4. Is the manuscript presented in an intelligible fashion and written in standard English?

Reviewer #1: Yes

Reviewer #2: Yes

5. Review Comments to the Author

Reviewer #1: There are some inconsistencies in the results and discussion that are critical to have fixed prior to acceptance of this manuscript.

February 11, 2022

Comments to the authors.

Overall, this manuscript is well written and adds significant information to previously published literature for regional limb perfusion in horses.

There are a few introduction, results, and discussion points that could be expanded on. The most critical discussion point and figure results that need to be clarified are also critical.

In several areas in the manuscript it is confusing as to whether pneumatic tourniquets > 400 mm Hg achieved higher MICs than pneumatic tourniquets <400 mm Hg. I have noted all areas that I found a discrepancy. However the entire manuscript should be reviewed to ensure this is clarified.

Lines 298- 299 Should read tourniquest <400 mm Hg pressures instead of > 400 mm Hg pressure

Lines 310- 316. Can the authors also discuss other possibilities why local anesthesia could enhance the effectiveness of the regional limb perfusion if any.

Lines 324- 325. What time period is 10x> MIC typically reached in regional limb perfusion. For example it might say.. Typically the CMAX:MIC>10 is reached by 15- 20 minutes. However, the mean time to peak concentrations are at 29 minutes.

Lines 331 to 339 You might also include.. This may allow the clinician to start proximal with the regional limb perfusion and on sequential treatments go more distal if there is a problem with a previous regional limb perfusion site.

Line 348 I would state... Administering 1/3 the systemic dose can be considered efficacious... The inclusion of no more than 1/3 the systemic dose leaves the authors open to “what about < 2%...

Figures S1 and S2 appear to be mislabeled where the Pneumatic tourniquet > 400 should be listed second, below the rubber tourniquet with the higher MIC. Beneath that the tourniquet < 400 should be listed third below the rubber tourniquet with the lowest MIC. A small label at the top of S1 and S2 stating susceptible bacteria and resistant bacteria would be helpful.

Please clarify what the significance is in including Table S2 and S3 for the readers as a modified table.

Both Figures and Tables appear the same. Probably they should all be labeled either Figures or Tables.

Reviewer #2: I have four comments:

1. Using the word "trial" for a study arm was a little confusing because trial is commonly used in a clinical trial, which is a study. Trial is also used in veterinary gait analyses. In line 154, it appears that trial and study are defined somewhat interchangeably. I think it would help your readers to change trial to arm or group. For what it's worth, Wikipedia's entry for meta-regression uses the word arm.

2. It is not necessary to include the univariate regression analysis. The multivariate regression is superior to the univariate one, so the univariate analysis adds nothing to the article, and that's especially true when the two analyses give different results. As it stands, it appears that readers have to make up their own minds about which analysis to believe. However, if the univariate truly adds something to the understanding of the data, then keep it in, but please explain why.

3. The forest plots caused me the most difficulty.

4. I understand the forest plots are used to visualize the meta-regression results but are effect sizes on the forest plot from the multiple regression, univariate regression, or raw data?

There are several things on the plots that I wasn't sure about:

1. The heading on the 4th column is "effect size with 95% CI," but effect size isn't mentioned in the text. How is it defined?

2. Please describe:

3. what Qb(1) means

4. what do all the p-values in the last column mean. What is being tested?

5. What is Q(121)? Is 121 correct, or should it be Q(122)?

6. There appear to be three heterogeneity statistics, but I think only one is described in the manuscript. Please define them in the statistics section. (Or just keep one of them.)

7. In line 245, you state, "Finally, meta-analysis showed that synovial structure, % systemic dose, tourniquet location, general anesthesia vs. standing, and tourniquet time were not statistically significantly associated with the outcome (Figs 4 and 5).

But in figure 5, tourniquet time has a p=0.02. Why is that not statistically significant?

That raises a larger question. I didn't see a cutoff for statistical significance in the manuscript. The contemporary approach in a study like this would be to call it an exploratory study and omit a cutoff and any mention of statistical significance. The p-values would be used like effects sizes to find the best therapy. Alternatively, you could do some kind of Type I error inflation correction and then use that method to give an overall error rate of 5%.

But as the analysis stands, with a 0.05 cutoff for each test, some of your results may be false-positive discoveries.

***if heterogeneous, how do you justify the regression?

6. PLOS authors have the option to publish the peer review history of their article (what does this mean?). If published, this will include your full peer review and any attached files.

Reviewer #1: No

Reviewer #2: No

---

## [Author Response · Author response to Decision Letter 0]

2 Mar 2022

Comments to the authors.

Overall, this manuscript is well written and adds significant information to previously published literature for regional limb perfusion in horses.

There are a few introduction, results, and discussion points that could be expanded on. The most critical discussion point and figure results that need to be clarified are also critical.

In several areas in the manuscript it is confusing as to whether pneumatic tourniquets > 400 mm Hg achieved higher MICs than pneumatic tourniquets <400 mm Hg. I have noted all areas that I found a discrepancy. However the entire manuscript should be reviewed to ensure this is clarified.

- The authors thank the reviewers for their comments and suggestions for the article. We believe we have addressed each issue and that the manuscript is improved thanks to the careful review.

Lines 298- 299 Should read tourniquets <400 mm Hg pressures instead of > 400 mm Hg pressure

- Based on our analysis, tourniquets with a pressure > 400mm Hg were actually the least successful in achieving Cmax:MIC ≥ 10. We were surprised by this result as we thought higher pressure tourniquets would be more effective, however, after review of the data we found that this was not the case.

Lines 310- 316. Can the authors also discuss other possibilities why local anesthesia could enhance the effectiveness of the regional limb perfusion if any.

- Thank you for your comment. We have added the following to the discussion: “Errico et al. (2008), based on preliminary studies, also suggested that weight shifting due to tourniquet associated pain leads to doubling of the intravascular pressure distal to the tourniquet which can lead to systemic leakage.” We do not have any evidence for other mechanisms of action. (Line 345-348)

Lines 324- 325. What time period is 10x> MIC typically reached in regional limb perfusion. For example it might say.. Typically the CMAX:MIC>10 is reached by 15- 20 minutes. However, the mean time to peak concentrations are at 29 minutes.

- Unfortunately, only 16 trials in 5 studies examined synovial fluid antibiotic concentration at time < 30 minutes. There were 2 trials that examined concentrations at 10 minutes, 12 that examined concentrations at 20 minutes, and 2 that examined concentrations at 25 minutes. Of these trials, 12/16 achieved Cmax:MIC ≥ 10 for sensitive bacteria and 6/16 achieved Cmax:MIC ≥ 10 for resistant bacteria. This information has been added to the discussion. (Line 358-362)

Lines 331 to 339 You might also include.. This may allow the clinician to start proximal with the regional limb perfusion and on sequential treatments go more distal if there is a problem with a previous regional limb perfusion site.

- Thank you for the comment. We think this is a very important point and added the following to the discussion: “Additionally, this may allow clinicians to begin regional limb perfusions in a more proximal location allowing for more distal locations in the vein to be used in sequential treatments if there are complications such as phlebitis”. (Line 377-380)

Line 348 I would state... Administering 1/3 the systemic dose can be considered efficacious... The inclusion of no more than 1/3 the systemic dose leaves the authors open to “what about < 2%...

- This sentence has been changed as suggested.

Figures S1 and S2 appear to be mislabeled where the Pneumatic tourniquet > 400 should be listed second, below the rubber tourniquet with the higher MIC. Beneath that the tourniquet < 400 should be listed third below the rubber tourniquet with the lowest MIC. A small label at the top of S1 and S2 stating susceptible bacteria and resistant bacteria would be helpful.

- Based on our analysis, tourniquets with a pressure > 400mm Hg were actually the least successful in achieving Cmax:MIC. We were surprised by this result as we thought higher pressure tourniquets would be more effective, however, after review of the data we found that this was not the case.

Please clarify what the significance is in including Table S2 and S3 for the readers as a modified table.

- The value is in providing full transparency and enabling other authors to repeat our analyses should they wish to. It is true that the full tables can be accessed directly from the primary paper, but we believed including them as supplementary material was in the spirit of promoting accessibility and reproducibility. If the reviewer believes them to be superfluous, we are fine to remove them. 

Both Figures and Tables appear the same. Probably they should all be labeled either Figures or Tables.

- We have separated figures and tables based off the appearance they have. We are happy to defer to the editors on this issue.

Reviewer #2: I have four comments:

1. Using the word "trial" for a study arm was a little confusing because trial is commonly used in a clinical trial, which is a study. Trial is also used in veterinary gait analyses. In line 154, it appears that trial and study are defined somewhat interchangeably. I think it would help your readers to change trial to arm or group. For what it's worth, Wikipedia's entry for meta-regression uses the word arm.

- Thank you for your suggestion. We agree and are happy to change “trial” to “arm”. 

2. It is not necessary to include the univariate regression analysis. The multivariate regression is superior to the univariate one, so the univariate analysis adds nothing to the article, and that's especially true when the two analyses give different results. As it stands, it appears that readers have to make up their own minds about which analysis to believe. However, if the univariate truly adds something to the understanding of the data, then keep it in, but please explain why.

- The authors feel that that univariate analysis is useful for clinicians and readers alike to see a graphic representation of the data so that they can assess Cmax:MIC for each variable even if differences in the variable do not significantly impact the outcome. If we only focus on the multivariate analysis, it is difficult for readers to, for example, look at our analysis and say “Ah yes, tourniquet time was not significant but all times did achieve Cmax:MIC>10”. 

3 and 4. The forest plots caused me the most difficulty. I understand the forest plots are used to visualize the meta-regression results but are effect sizes on the forest plot from the multiple regression, univariate regression, or raw data?

- We agree that the forest plots are difficult to read and interpret simply because there are so many arms included in all of the studies, but we believe it best practice to include these raw data in at least some form in the manuscript. What is labeled “Effect size” should actually be “multiples of MIC” – it just happens that the default label for our software is “Effect size”. We have changed this label to “multiples of MIC” to clarify. 

There are several things on the plots that I wasn't sure about:

1. The heading on the 4th column is "effect size with 95% CI," but effect size isn't mentioned in the text. How is it defined?

- See previous comment about labels.

2. Please describe what Qb(1) means.

- Qb is a test of the group differences within the subgroup on univariable analysis. We have removed it from the figures to avoid confusion. 

3. What do all the p-values in the last column mean. What is being tested?

- Thank you for bringing this to our attention. These are default outputs from the software, which is expecting an effect size, and the p-value is testing the hypothesis that the coefficient is significantly different from one. Since this hypothesis testing is not relevant in this case, we have removed it from the figures. 

5. What is Q(121)? Is 121 correct, or should it be Q(122)?

- This is Cochran’s Q, which is a measure of heterogeneity. 121 is correct, because there was one arm for which we were not able to get measures of dispersion, so the total number of arms contributing to the meta-analysis is 122 (you’ll see in the Forest plot of all studies that “Study 13” is missing). But thank you for catching this – we had missed it! We have added this explanation to the Results section. 

6. There appear to be three heterogeneity statistics, but I think only one is described in the manuscript. Please define them in the statistics section. (Or just keep one of them.)

- We have decided to keep only the I2 and Cochran Q tests. We have added explanation to the methods and removed the other statistics from the figures.

7. In line 245, you state, "Finally, meta-analysis showed that synovial structure, % systemic dose, tourniquet location, general anesthesia vs. standing, and tourniquet time were not statistically significantly associated with the outcome (Figs 4 and 5).

But in figure 5, tourniquet time has a p=0.02. Why is that not statistically significant?

- The test of group differences in this figure shows results of univariable analysis. To avoid confusion, we have removed the Qb and its p-value from the figure, since the p-value of the multivariable meta-regression was the one that we considered of most importance. We also removed the p-values from the right hand column, since these p-values had to do with the default output of the software, which is an effect size (OR or RR). 

That raises a larger question. I didn't see a cutoff for statistical significance in the manuscript. The contemporary approach in a study like this would be to call it an exploratory study and omit a cutoff and any mention of statistical significance. The p-values would be used like effects sizes to find the best therapy. Alternatively, you could do some kind of Type I error inflation correction and then use that method to give an overall error rate of 5%.

But as the analysis stands, with a 0.05 cutoff for each test, some of your results may be false-positive discoveries.

- We agree that a Type 1 error inflation correction is warranted at this point, and we have used a Bonferroni-corrected p-value of 0.006 (0.05 divided by the 8 independent variables being tested). The two variables that were found to be statistically significant with the uncorrected p-value remain so with the corrected p-value. 

***if heterogeneous, how do you justify the regression?

- As indicated by the Cochran Review, the meta-regression is used to explore sources of heterogeneity, not to obtain a summary measure despite heterogeneity. Therefore, we believe the regression is appropriate here.

---

## [Editor Report · Decision Letter 1]

11 Mar 2022

Effects of regional limb perfusion technique on concentrations of antibiotic achieved at the target site: a meta-analysis

PONE-D-22-00403R1

Dear Dr. Ortved,

We’re pleased to inform you that your manuscript has been judged scientifically suitable for publication and will be formally accepted for publication once it meets all outstanding technical requirements.

Kind regards,

Richard Evans

Academic Editor

PLOS ONE

Additional Editor Comments (optional):

Thank you for responding promptly to our comments. One last thing to double-check: Near the bottom of your responses, you state, "we have used a Bonferroni-corrected p-value of 0.006 (0.05 divided by the 8 independent variables being tested)." However, the Bonferroni correction should be 0.05 divided by the number of statistical tests, not the number of independent variables. I'm assuming that in your case, those two adjustments are the same. Please check that your inferences don't change using the correct Bonferroni correction.

---

## [Editor Report · Acceptance letter]

16 Mar 2022

PONE-D-22-00403R1 

Effects of regional limb perfusion technique on concentrations of antibiotic achieved at the target site: a meta-analysis 

Dear Dr. Ortved:

I'm pleased to inform you that your manuscript has been deemed suitable for publication in PLOS ONE. Congratulations! Your manuscript is now with our production department. 

Kind regards, 

on behalf of

Dr. Richard Evans 

Academic Editor

PLOS ONE